# Light sources with bias tunable spectrum based on van der Waals interface transistors

Hugo Henck [1,2], Diego Mauro[1,2], Daniil Domaretskiy [1,2], Marc Philippi[1,2], Shahriar Memaran[3,4], Wenkai Zheng[3,4], Zhengguang Lu[3,4], Dmitry Shcherbakov [5], Chun Ning Lau[5], Dmitry Smirnov [3,4], Luis Balicas [3,4], Kenji Watanabe [6], Takashi Taniguchi [7], Vladimir I. Fal'ko [8,9], Ignacio Gutiérrez-Lezama[1,2], Nicolas Ubrig [1,2✉] & Alberto F. Morpurgo [1,2✉]

Light-emitting electronic devices are ubiquitous in key areas of current technology, such as data communications, solid-state lighting, displays, and optical interconnects. Controlling the spectrum of the emitted light electrically, by simply acting on the device bias conditions, is an important goal with potential technological repercussions. However, identifying a material platform enabling broad electrical tuning of the spectrum of electroluminescent devices remains challenging. Here, we propose light-emitting field-effect transistors based on van der Waals interfaces of atomically thin semiconductors as a promising class of devices to achieve this goal. We demonstrate that large spectral changes in room-temperature electroluminescence can be controlled both at the device assembly stage –by suitably selecting the material forming the interfaces– and on-chip, by changing the bias to modify the device operation point. Even though the precise relation between device bias and kinetics of the radiative transitions remains to be understood, our experiments show that the physical mechanism responsible for light emission is robust, making these devices compatible with simple large areas device production methods.

[1] Department of Quantum Matter Physics, University of Geneva, 24 Quai Ernest Ansermet, 1211 Geneva, Switzerland. [2] Department of Applied Physics, University of Geneva, 24 Quai Ernest Ansermet, 1211 Geneva, Switzerland. [3] National High Magnetic Field Laboratory, Tallahassee, FL 32310, USA. [4] Department of Physics, Florida State University, Tallahassee, FL 32306-4350, USA. [5] Department of Physics, The Ohio State University, Columbus, OH 43210, USA. [6] Research Center for Functional Materials, National Institute for Materials Science, 1-1 Namiki, Tsukuba 305-0044, Japan. [7] International Center for Materials Nanoarchitectonics, National Institute for Materials Science, 1-1 Namiki, Tsukuba 305-0044, Japan. [8] National Graphene Institute, University of Manchester, Booth Street East, M13 9PL Manchester, UK. [9] Henry Royce Institute for Advanced Materials, M13 9PL Manchester, UK. ✉email: nicolas.ubrig@unige.ch; alberto.morpurgo@unige.ch

Since the discovery that monolayer semiconducting transition metal dichalcogenides (TMDs) are direct gap semiconductors exhibiting strong luminescence[1,2], two-dimensional (2D) materials have attracted interest for optoelectronic applications[3–6]. Their potential stems from the ease with which the electronic properties of 2D semiconductors can be tuned by different means[7–11]. In phosphorene, for instance, mechanical strain can be used to tune the bandgap by a very large amount, resulting in a controllable change of the wavelength of emitted light, as observed in recent photoluminescence measurements[12]. In semiconducting TMDs, electrostatic gating gives access to a variety of excitonic states with different energy[13,14], an effect that has been used to realize bias-tunable electroluminescent devices with emission energy that has been varied by several tens of meV[9,11]. Realizing electroluminescent devices enabling much broader changes in the spectrum of the emitted light by simply acting on the device operation point (i.e., on the device bias) has however not been possible so far.

Light-emitting field-effect transistors (LEFETs) are three-terminal devices that allow switching of both the electrical conductance and light emission[15–19]. They rely on semiconductors that support ambipolar transport to inject simultaneously in the transistor channel electrons and holes[20], whose radiative recombination is the origin of the emitted light[21,22]. Past research on LEFETs has concentrated on organic semiconductors, which have suitable properties for their realization[18,19,23–25]. Ionic-gated LEFETs based on 2D semiconductors are a recently discovered alternative that offer potential advantages, such as higher and well-balanced electron and hole mobilities, as well as low-bias operation[26–30]. Efficient LEFETs, however, require the use of 2D semiconductors with a direct bandgap, whose paucity limits the possibility to tune the spectrum of the emitted light. Van der Waals (vdW) interfaces formed by atomically thin semiconducting materials provide a strategy to address this issue because the wavelength of light emitted by interlayer transitions (electrons hosted in one layer recombining with holes hosted in the other) can be engineered by selecting constituent materials with an appropriate band alignment[31–35].

Here, we demonstrate experimentally LEFETs realized on vdW interfaces, and show that they can be operated as electrically tunable light sources. As compared to LEFETs based on individual monolayers, devices fabricated on vdW interfaces potentially offer more functionality. The electronic structure of the individual layers, for instance, is often only minorly affected by the interface formation, so that a rich set of electronic levels—i.e., the bands of the two materials, including the sub-bands originating from quantum confinement- is present[36–40]. If properly populated by acting on the device operation point (i.e., the applied source-drain and gate voltages, $V_{SD}$ and $V_G$, respectively), these levels may enable the energy of the emitted light to be tuned. Additionally, the electric field perpendicular to the transistor channel creates a potential difference between the layers forming the interface, which shifts the energy of the recombining electrons and holes[41,42]. The wavelength of light generated by interlayer transitions is expected to shift accordingly, providing another route to tune the emission spectrum by acting on the device operation point. These ideas disclose possible mechanisms to operate LEFETs based on vdW interfaces as electrically tunable light sources. However, neither their validity nor the potential of LEFETs based on vdW interfaces has been assessed so far.

Here, we show that light-emitting transistors based on a recently discovered type of van der Waals (vdW) heterostructures —which we refer to as Γ–Γ interfaces (see discussion below)— exhibit room-temperature electroluminescence that can be tuned over a much broader spectral range (from below 1.2 to 1.7 eV, in the devices reported here) by acting exclusively on the device operation point. Our work relies on devices made of bilayers (2L) of semiconducting transition metal dichalcogenides (TMDs; we use $WS_2$ and $MoS_2$) and InSe multilayers, to form vdW interfaces that belong to a recently identified class exhibiting robust radiative interlayer transitions (e.g., transitions that are radiative irrespective of the lattice structure of the constituent materials or of their relative orientation)[38,43,44]. The robustness originates from having the conduction and valence band extrema in the two layers at $k=0$, i.e., at the Γ-point of the Brillouin zone (which is why we refer to these systems as Γ–Γ interfaces. It is important because it facilitates the device assembly, and makes it compatible with simple large-area production techniques[9,45]. Indeed, we find that all our LEFETs exhibit electroluminescence, with a wavelength that can be engineered by selecting the constituent layers, and with a spectrum that can be tuned by acting on the device operation point. Contrary to earlier studies of Γ–Γ interfaces[44]— in which photoluminescence (PL) was only observed at cryogenic temperatures– electroluminescence (EL) is already present at room temperature, a key finding when assessing the technological potential of these devices.

## Results

**Device fabrication and characterization.** Ionic-gated LEFETs (see Fig. 1a–c and Supplementary section 1) based on 2L-TMD/InSe vdW interfaces (see Fig. 1d) are realized using techniques commonly employed for the assembly of structures based on 2D materials[46]. TMD bilayers and InSe multilayers are exfoliated from bulk crystals onto Si/SiO$_2$ substrates. Heterostructures are formed by picking up layers one after the other, and transferring the resulting interface onto a fresh Si/SiO$_2$ substrate, with the TMD layer covering the InSe one and effectively encapsulating it (which is important in view of the non-perfect stability of InSe in ambient; the interface assembly process is carried out in the controlled atmosphere of a glove box). Source and drain contacts, as well as a large pad acting as gate electrode, are defined by means of electron-beam lithography, electron-beam evaporation of a Pt-Au film (5/30 nm) and lift-off (see Fig. 1e). Subsequently, a *window* in PMMA is patterned to define the region where the ionic liquid ((N,N-diethyl-N-methyl-N-(2-methoxyethyl) ammonium bis(trifluoromethylsulfonyl) imide) commonly referred to as DEME-TFSI) contacts the interface. The liquid is applied as a final step, prior to inserting the devices in a vacuum chamber with optical access (see "Methods" section for more details).

The $WS_2$ and $MoS_2$ crystals used for exfoliation are purchased from HQ Graphene. InSe is a less commonly employed compound and care is needed because the crystal quality varies strongly depending on details of the growth process. Lower quality can result in a large density of defects[47–49] that create in-gap states acting as hole traps, which is why our earlier attempts to use InSe crystals to realize vdW interface LEFETs failed (in-gap states prevent the electrostatic accumulation of holes in the valence band of the TMD, and impede ambipolar transport). The InSe crystals that we employ here, grown at Florida State University (see "Methods" section for details of the growth process), appear to have high quality and a low density of defects, as we directly infer from the transistor electrical characteristics and from the narrow lines observed in PL studies of h-BN-encapsulated multilayers (see Fig. 1f).

Figure 2a shows the room-temperature transfer curve (source-drain current $I_{SD}$ vs. gate voltage $V_G$ at fixed source-drain voltage $V_{SD} = 50$ mV) of a device realized on a 2L-WS$_2$/4L-InSe interface. The behavior is typical of ambipolar transistors with current mediated by holes and electrons flowing for sufficiently large negative and positive $V_G$, respectively[24,26,50–55]. Accumulation of electrons and holes leads to comparable current levels, confirming

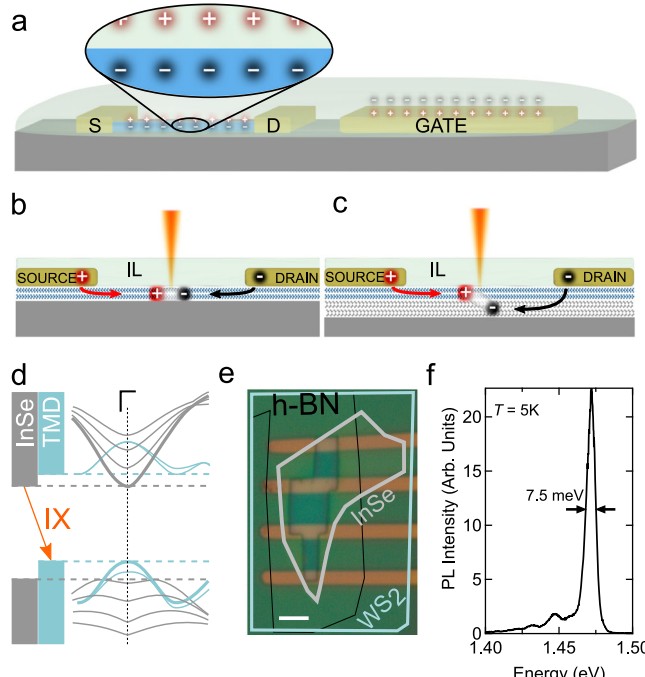

**Fig. 1 LEFETs based on van der Waals interfaces. a** Schematics of an ionic-gated field-effect transistor, with the source-drain (S-D) contacts connected to a semiconducting layer and the gate electrode, all in contact with an ionic liquid. The zoom in on the channel regions shows that upon the application of a gate voltage charge is accumulated on the semiconductor (see Supplementary Section 1 for details). **b**, **c** Electron-hole recombination in the channel of an ionic liquid (IL)-based LEFET operated in the ambipolar injection regime (with electrons and holes injected at opposite contacts) for a device based on an individual 2D material (**b**) and on a vdW interface (**c**). In the latter case, light is emitted by the recombination of electrons and holes hosted in different layers, which offers new opportunity to control its spectrum (see main text; here we use interfaces of bilayer TMDs, hosting holes, and InSe multilayers, hosting electrons). **d** The type-II band alignment between InSe and TMDs around the Γ-point enables k-direct interlayer radiative transitions between the two band edges (as indicated by the orange arrow and labeled IX). Higher energy bands originating from the quantum confinement of charge carriers in the two layers (as indicated by the thin lines) can lead to radiative transitions with different energy. **e** Optical micrograph of a device used in this work, based on 2L-WS$_2$ and 4L InSe (the contours of the layers are marked by the blue and white lines; the stacking sequence is indicated on top). The vdW interface is in contact with the liquid through an opening in the PMMA layer covering the entire sample. The scale bar is 2 μm. **f** Peak in the PL spectrum of a 5L InSe crystal encapsulated between two thicker hBN layers: the narrow width indicates the high quality of the material.

that transport is well-balanced and that residual defects in InSe do not prevent high-quality device operation. When plotted in logarithmic scale (Fig. 2b) the data allow determining the subthreshold swing $S = \ln 10 \frac{dV_G}{d(\ln I_{SD})}$, equal to $S = 115$ and 90 mV/decade near the threshold for hole and electron conduction, respectively (other devices exhibit values even closer to the ultimate room-temperature limit of 60 mV/decade)[56]. The output curves ($I_{SD}$-vs.-$V_{SD}$ plotted for different, fixed $V_G$) shown in Fig. 2c, d also exhibit the expected behavior. Upon increasing $V_{SD}$, $I_{SD}$ increases linearly at first then saturates, and eventually exhibits a very steep increase, when entering the ambipolar injection regime. This regime—in which electrons and holes are injected at opposite contacts—is the one of interest for LEFET

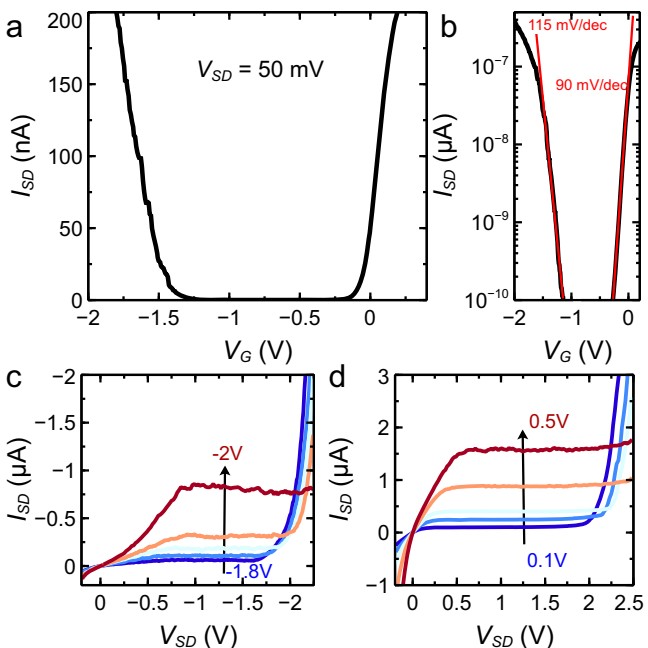

**Fig. 2 Transistor characteristics of a 2L-WS$_2$/4L InSe device. a** Room-temperature transfer curve ($I_{SD}$-vs.-$V_G$ at fixed $V_{SD} = 50$ mV) of a 2L-WS$_2$/4L-Inse. **b** The high device quality is evidenced by the low subthreshold swings for electron and hole transport, whose values (115 mV/decade and 90 mV/decade, respectively, for this specific transistor) are close to the ultimate limit of 60 mV/decade. **c** and **d** Output characteristics ($I_{SD}$ as function of $V_{SD}$) of the same device for different negative and positive gate biases, respectively. Linear, saturation, and ambipolar injection regimes can be clearly identified upon increasing the magnitude of $V_{SD}$, as described in the main text (see also Supplementary section 1).

operation (see Supplementary section 1), and can be reached irrespective of the polarity of the applied gate voltage.

**Electroluminescence from vdW interface LEFET.** EL is expected to occur concomitantly with ambipolar injection, with light emission starting at one of the contacts and shifting into the channel as $V_{SD}$ is further increased[18]. This is indeed what we observe (see Fig. 3a as well as Supplementary Fig. 2 and accompanying discussion in Supplementary section 1). The light emitted by the LEFET is collected by a microscope objective and fed into a spectrometer. The spectral analysis performed on data measured at fixed $V_G$, by increasing $V_{SD}$ past the onset of the ambipolar injection regime ($V_{SD} > +1.9$ V in Fig. 3b for $V_G > 0$ and $V_{SD} < -2.2$ V in Fig. 3e for $V_G < 0$ V), is shown in Fig. 3c and d for $V_G > 0$ V, and in Fig. 3f, g for $V_G < 0$ V. As the current increases exponentially rapidly, we initially limit the maximum applied $V_{SD}$ to avoid damaging the devices. EL is detected in all cases, with an intensity that increases rapidly with $V_{SD}$. The light exhibits a dominant spectral line just above 1.2 eV, irrespective of the precise value of $V_{SD}$ and of whether $V_G$ has positive or negative polarity (an additional shoulder at 1.4 eV is visible for $V_G < 0$, see Fig. 3g, is also present—albeit less pronounced—for $V_G > 0$).

In Fig. 3h, we plot the device EL spectrum (red line; $V_G = +0.5$ V and $V_{SD} = +2.2$ V) together with the PL spectrum of 4L-InSe (purple line; PL is measured at $T = 5$ K, since no signal is observed at room temperature) and 2L-WS$_2$ (blue line). The energy of the EL peak is considerably lower than the recombination energy in either 4L-InSe or 2L-WS$_2$, as expected for an interlayer transition[44]. The energy of the room-temperature EL signal (Fig. 3j, thick line) matches that of low-temperature

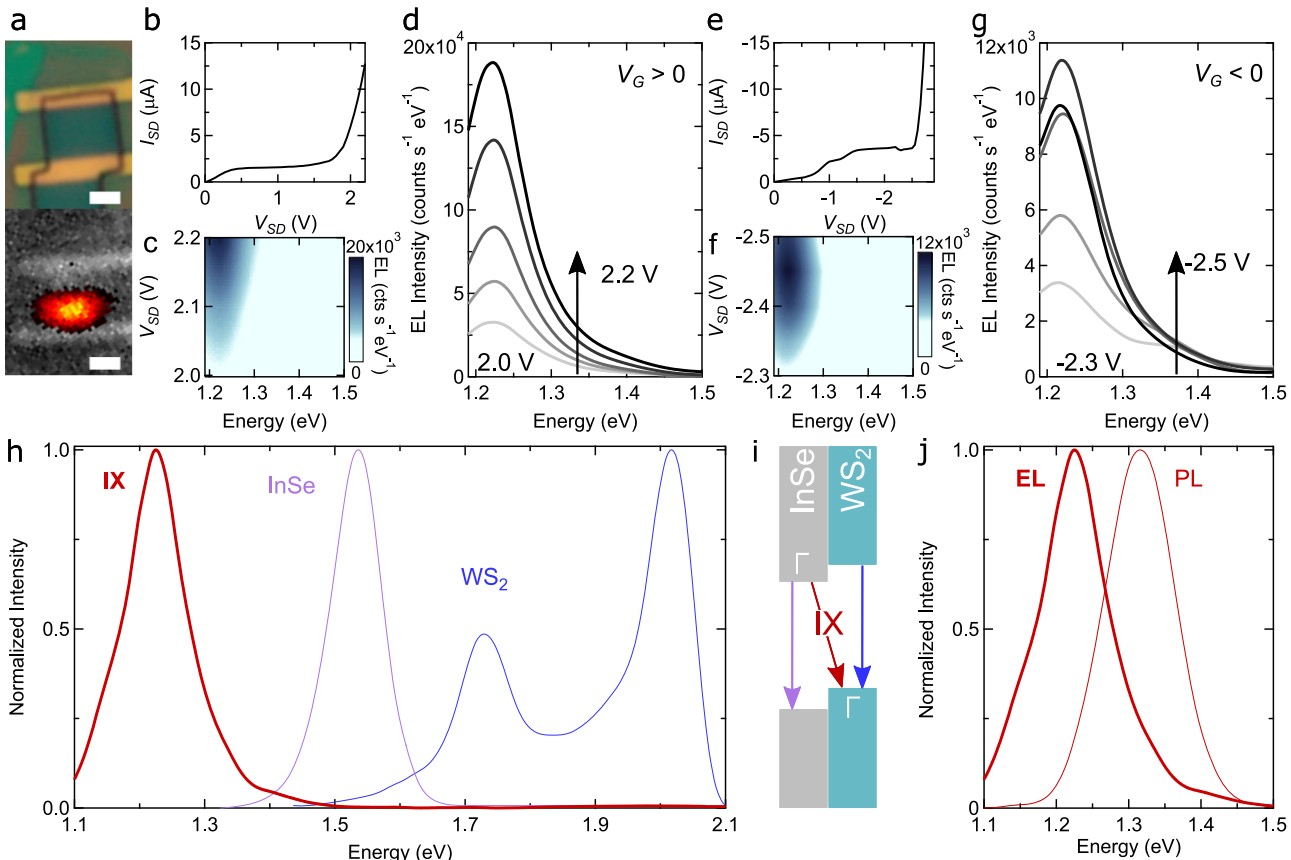

**Fig. 3 Electroluminescence from a Γ–Γ interface. a** Images of the channel of a transistor based on a 2L-WS$_2$/4L-InSe heterostructure taken with an optical microscope (top) and with the camera of our spectrometer (bottom). The bottom image, taken with the device biased at the onset of the ambipolar injection regime, shows a bright spot due to EL. The scale bars are 1 μm. **b** Device output curve measured at, $V_G = +0.5$ V (electron accumulation). **c** False-color plot of the EL intensity measured by the spectrometer, as a function of photon energy and applied $V_{SD}$, showing a peak centered around 1.25 eV that emerges at $V_{SD} = +1.9$ V, corresponding to the onset of ambipolar injection (see **b**). **d** Individual EL spectra at selected $V_{SD}$ values extracted from **c**: in this bias range the spectrum remains unchanged, and the intensity increases following the increase in source-drain current. **e–g** Data analogous to those of panels **b–d** are shown for $V_G = -2.2$ V (hole accumulation), demonstrating that the presence of EL is robust and that at sufficiently low bias the spectrum is virtually identical for electron and hole accumulation. **h** The peak in the EL spectrum (acquired with $V_{SD} = +2.2$ V and $V_G = +0.5$ V) is red-shifted relative to the PL emission energies of the layers forming the interface (purple: 4L InSe; blue: 2L-WS$_2$; for InSe, PL data are taken at 5 K, because no PL is observed at room temperature), as expected from an interlayer Γ–Γ transition (see **i**). **j** Comparison between the normalized EL (thick red line) and PL (thin red line) emission spectra of the interface. The energy difference originates from having to measure PL at cryogenic temperatures ($T = 5$ K), since no PL is observed at room temperature.

interlayer transitions seen in PL (Fig. 3j, thin line) due to electrons in InSe recombining with holes in WS$_2$, if we take into account that the TMD and the InSe gap typically increases by approximately 50-100 meV upon cooling from 300 to 5 K[14,57–59]. The measurements, therefore, confirm that our LEFET operates as anticipated, with electrons injected in the InSe layer and holes in the WS$_2$ one recombining via an interlayer transition. Finding that this transition results in EL even at room temperature is a positive, unexpected surprise.

Devices based on other Γ–Γ vdW interfaces should exhibit all key properties of 2L-WS$_2$/4L-InSe LEFETs. We verify that this is indeed the case using transistors realized on interfaces of 2L-MoS$_2$ (instead of 2L-WS$_2$), and 3L-, 4L-, and 5L-InSe. Without going through all details (see Supplementary section 2), the data show the occurrence of ambipolar transport (Fig. 4a) and of the ambipolar injection regime past saturation (Fig. 4b). Upon entering the ambipolar injection regime, EL is observed resulting in a line at 1.3 eV (for 2L-MoS$_2$/5L-InSe) independently of $V_{SD}$ (see Fig. 4c, d), i.e., an energy lower than that of the transitions in the constituent materials (see Fig. 4e). Figure 4f overviews the results obtained, by plotting together the room-temperature EL

spectrum of LEFETs fabricated on all different vdW interfaces, and shows that combining different 2D materials indeed allows a dense coverage of part of the near-infrared and visible spectral range. Selecting multilayers of different thicknesses or having different compositions (e.g., MoSe$_2$ or MoTe$_2$) would further broaden the accessible spectrum, both on the higher and lower end[38,43,44].

**Electrically tunable EL spectrum in vdW interface LEFET.** Having established that LEFETs based on Γ–Γ interfaces provide a robust platform to generate room-temperature EL, we test whether the light spectrum can be controlled by varying the device operation point. Figure 5 illustrates the evolution of the spectrum of the light emitted by a 2L-WS$_2$/4L-InSe (Fig. 5a–c) and by a 2L-MoS$_2$/5L-InSe (Fig. 5d–f) LEFETs, upon pushing the source-drain bias $V_{SD}$ to reach deeper in the ambipolar transport regime. The green rectangle in Fig. 5a delimits the $V_{SD}$ interval discussed earlier, and the corresponding part of the spectrum in Fig. 5c (also delimited by a green rectangle) shows emission from the interlayer transition just above 1.2 eV, in agreement with the data

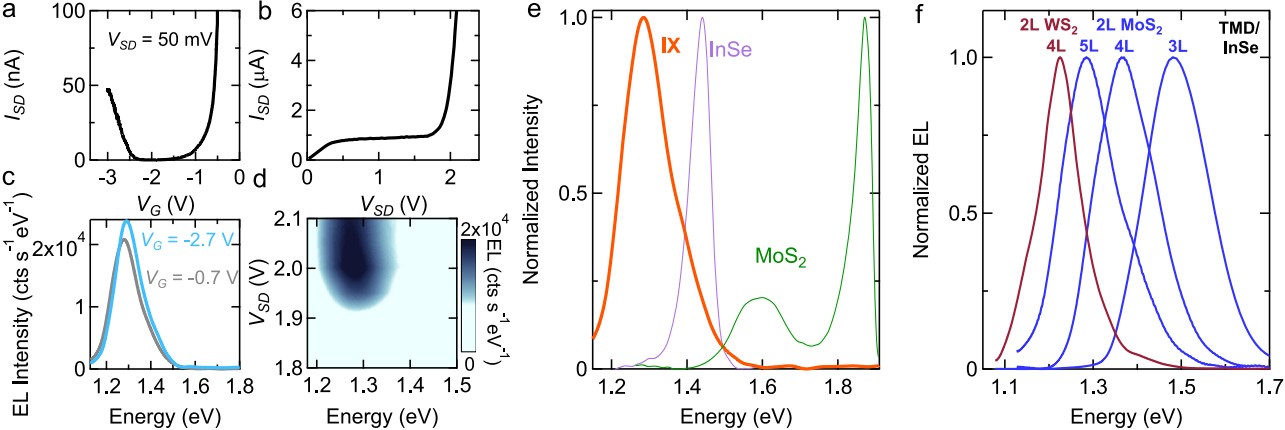

**Fig. 4 LEFET based on a MoS₂/InSe interface. a, b** Transfer and output characteristics of a 2L-MoS₂/5L-InSe transistor. **c** EL spectra of the device biased near the onset of ambipolar injection, for electron and hole accumulation (the gray and blue curves are measured respectively at $V_G = -0.7$ V and $V_{SD} = +2.1$ V, and at $V_G = -2.7$ V and $V_{SD} = -2.3$ V), showing a peak around 1.3 eV independently of the applied gate voltage. **d** Dependence of EL intensity on photon energy and $V_{SD}$, measured at $V_G = -0.7$ V, showing that near the onset of ambipolar transport the spectrum is independent of $V_{SD}$. **e** Also in this case, the EL spectrum (thick orange line) is red-shifted as compared to the PL emission energy of the layers forming the interface (purple: 4L InSe; green: 2L-MoS₂), as expected for an interlayer transition. **f** EL spectrum of LEFETs realized using four different interfaces, based on two different 2L TMDs (blue: MoS₂; red: WS₂) and three different thicknesses of the InSe layer (3L, 4L, and 5L, as indicated in the figure), showing a dense coverage of part of the visible spectrum (a broader range of photon energy can be spanned using other semiconducting TMD compounds).

shown in Fig. 3b, c. When larger $V_{SD}$ is applied, corresponding to the interval in the rectangle delimited by the red line in Fig. 5a, the spectrum evolves. Additional transitions appear, visible in Fig. 5b in the region delimited by the red rectangle, as well as in Fig. 5c, which shows the spectrum of the emitted light at specific values of $V_{SD}$. A qualitatively identical behavior is observed in LEFETs based on 2L-MoS₂/5L-InSe, with the regime of lower and higher $V_{SD}$ highlighted by red and green rectangles in Fig. 5d and the corresponding spectra shown in Fig. 5e, f.

We have also measured the spectrum of the emitted light at a fixed $V_{SD}$ value, as a function of $V_G$, and found that in that case as well, the spectrum depends strongly on the device operation point. Figure 6a–d show the spectrum of the light emitted by a device realized on a 2L-WS₂/4L-InSe interface, as a function of gate voltage, for two different values of source-drain bias ($V_{SD} = +2.4$ V in Fig. 6a, b and $V_{SD} = +3.12$ V in Fig. 6c, d). Changing the gate voltage at fixed $V_{SD}$ allows switching the spectrum of the light between two transitions visible in Fig. 6b, c. In particular, at large positive gate voltage ($V_G > +0.4$ V in Fig. 6a and $V_G > +0.8$ V in Fig. 6c), the spectrum of the emitted light is dominated by the interlayer transition at 1.2 eV between the bottom of the InSe conduction band and the top of the TMD valence band. At low gate voltage ($V_G < +0.2$ V in Fig. 6a and $V_G < +0.6$ V in Fig. 6c), instead, light is emitted by another transition (possibly by multiple transitions, as suggested by the broad linewidth) at higher energy (approximately 1.4 eV), which appears to blue shift upon increasing $V_G$. Unexpectedly, the two regimes are separated by an interval of gate voltages in which the power of emitted light vanishes (or is below the sensitivity of our detector). Finding that the gate allows switching the spectrum between two different emission lines is interesting, as it may provide new functionality to these LEFETs devices.

As part of our experiments, we have also determined the external quantum efficiency of our devices—i.e., the ratio between the number of emitted photons detected in our set-up and the number of injected electrons—by comparing the measured EL signal to the signal measured (with the same set-up) when using a commercial light-emitting diode as source. We found that, in the experimental configuration employed to detect EL, the external quantum efficiency of our devices is approximately 0.005%, three

orders of magnitude smaller than that of commercial devices. It should be realized, however, that in our set-up the external quantum efficiency is much lower than the actual quantum efficiency, because the optical selection rules for interlayer transitions[37,60] dictate that light is mostly emitted in the plane of the interface. This implies that the majority of the photons emitted by our light-emitting transistors are not collected by the microscope objective, and that adopting strategies to improve the light outcoupling should lead to a drastic enhancement of the measured external quantum efficiency.

Irrespective of these considerations about the external quantum efficiency, our observation that the EL spectrum does depend on the device operation point proves that LEFETs based on vdW interfaces are indeed electrically tunable light sources. Understanding in detail how the EL spectrum depends on the LEFET operation point is however complex, both because different processes likely play a role, and because screening due to charges accumulated in the transistor channel can strongly (and non-linearly) affect the potential difference between the two layers forming the interface, especially in the region of the transistor channel where electron-hole recombination occurs. At sufficiently large $V_{SD}$, we expect that electrons are injected not only in the conduction band of InSe but also in that of the TMD, so that light can be emitted also from intralayer transitions within the TMD. This may account for the peak centered around 1.6 eV in 2L-WS₂/4L-InSe, which corresponds well to one of the 2L-WS₂ PL peaks. An intralayer transition in the semiconducting TMD is likely also responsible for part of the broad peak around 1.5 eV in the MoS₂-based interfaces (the energy matches one of the peaks observed in PL of 2L-MoS₂). The less pronounced peaks near 1.4 eV (at comparable but different energies in the 2L-WS₂ and the 2L-MoS₂ devices; see Fig. 5b, e) occur at an energy that changes slightly upon changing $V_{SD}$. As the energy of these peaks is lower than all known intralayer transitions in the respective systems, we attribute their origin to an interlayer transition between an electron in a higher energy InSe sub-band recombining with a hole in the TMD (whose precise energy is affected by the electrostatic potential difference between the layers). This attribution is also consistent with data taken at fixed $V_{SD}$ upon varying $V_G$ (see Fig. 6), in which the transition energy is seen to blue shift upon changing the gate voltage.

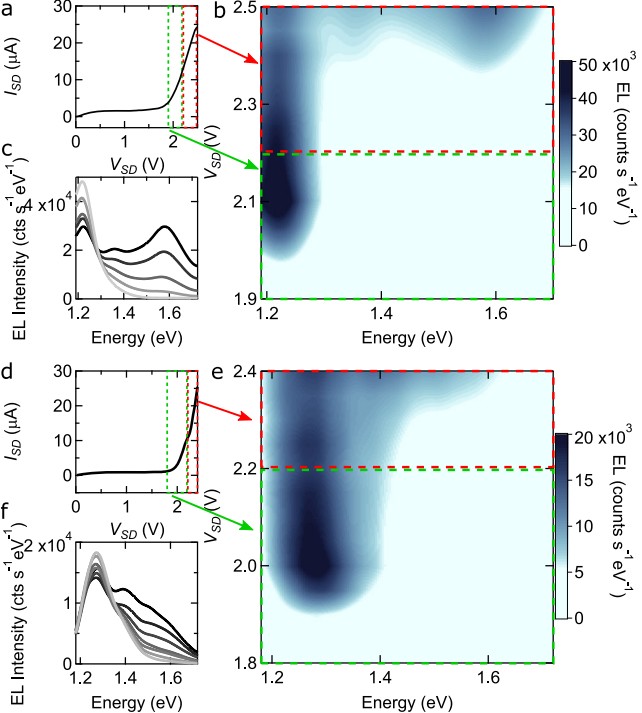

**Fig. 5 Bias-tunable light emission from vdW interface LEFETs. a** Output characteristics of the 2L-WS$_2$/4L-InSe device, whose data are shown in Fig. 3 ($V_G$ = +0.2 V). The colored dashed lines delimit the low-bias (green line) and high-bias (red line) regime, which exhibit different EL spectral properties. **b** Color plot of the EL spectrum as a function of photon energy and $V_{SD}$. In the low-bias regime (region inside the green rectangle) EL exhibit a single peak, due to a Γ–Γ interlayer transition from the bottom of the InSe conduction band to the top of the WS$_2$ valence band (see Fig. 3). In this regime the spectrum is independent of bias. In the high-bias regime (region inside the red rectangle), the EL spectrum evolves upon increasing $V_{SD}$, showing that the LEFET acts a light source with bias-tunable spectrum. **c** Individual EL emission spectra measured in the high-bias regime, upon varying $V_{SD}$ from 2.3 to 2.5 V. **d–f** Same measurements as those shown in panels **a–c**, but performed on a 2L-MoS$_2$/5L-InSe LEFET (data taken at $V_G$ = –0.7 V). The data illustrate that the evolution of the EL spectrum observed in this MoS$_2$/InSe LEFET is fully analogous that observed in the WS$_2$/InSe LEFET, showing the robustness of the device operation.

More work is clearly needed to understand in detail the electroluminescence spectrum of Γ–Γ interfaces at large biases, as well as its evolution with both source-drain and gate biases (as mentioned earlier, screening plays a major and complex role in the way spectral features shift as function of biases, and a quantitative description will require a separate, dedicated modeling effort). We note, however, that the spectrum of light emitted by LEFETs based on monolayer TMDs remains unchanged even under driving the device with very large source-drain biases, as discussed in Supplementary section 3 and shown in Supplementary Fig. 4 in there. The data, therefore, appear to substantiate our initial idea—namely that LEFETs based on vdW interfaces offer more functionalities than similar devices realized from individual monolayers—irrespective of the precise microscopic origin of the emitted light (i.e., of the specific transitions involved in the light emission process).

## Discussion

The results presented above demonstrate the operation of vdW interface LEFETs and show that these transistors do allow the realization of light sources with a bias-tunable spectrum. Finding that devices realized with multiple semiconducting TMDs and with InSe layers of different thickness lead to a qualitatively similar evolution of the spectrum of the emitted light with bias indicates that the operation mechanism is robust and of general validity. This robustness is important because—whereas only a few examples of electroluminescent devices with an electrically controllable spectrum have been reported in the past[61–64]—a large variety of 2D semiconductors exists that can be employed to realize light-emitting Γ–Γ interfaces.

It is clear that at this stage light-emitting transistors based on Γ–Γ vdW interfaces remain proof-of-principle devices and that considerable research is needed to characterize them and optimize their operation. We anticipate, for instance, that ionic liquid gating can be replaced by conventional solid-state gates using nm-thick h-BN dielectrics to separate the gate electrode from the vdW heterostructure. Indeed, h-BN layers that are a few nanometers thick exhibit breakdown field values approaching 1 V/nm[65], which are likely sufficient to achieve ambipolar transport and to operate the devices with $V_{SD}$ larger than $V_G$. Similarly—as already mentioned—the external quantum efficiency of our devices will need to be improved by adopting suitable strategies to optimize light outcoupling. While it is clear that at this stage

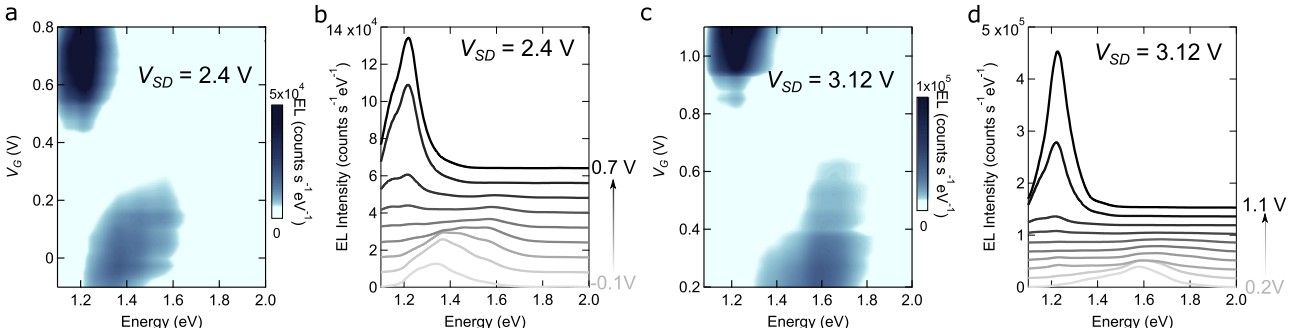

**Fig. 6 Gate-tuning of the spectrum of the light emitted from a 2L-WS$_2$/4L-InSe LEFET device. a** Color plot of the intensity of the light emitted by a 2L-WS$_2$/4L-InSe LEFET device as function of photon energy and gate voltage ($V_{SD}$ is fixed to +2.4 V). The data show that varying $V_G$ allows the frequency of the emitted light to be tuned: for $V_G$ > 0.4 V, the spectrum is peaked at 1.25 eV, i.e., the characteristic energy of the interlayer transition between the bottom of the InSe conduction band and the top of the WS$_2$ valence band; for $V_G$ < 0.2 V a broader peak centered around 1.4 eV and shifting with varying $V_G$ is observed. **b** Individual horizontal cuts of the color plot shown in **a**, for $V_G$ varying from –0.1 to +0.7 V in 0.1 V steps. **c** and **d** Same measurements as in **a** and **b** with $V_{SD}$ fixed at +3.12 V. The individual spectral cuts range from $V_G$ = +0.2 V to +1.1 V in 0.1 V steps.

different aspects of the device require improvements, it is worth re-iterating that Γ–Γ interfaces used within a light-emitting transistor configuration represent a platform that offers a very high potential for the realization of electroluminescent devices with bias-tunable spectrum, and that satisfies many key requirements essential to the development of a successful technology. These include room-temperature operation and insensitivity of the devices to details of their assembly process, which ensures their robust operation. It is for these reasons that exploring the development of light-emitting transistors based on Γ–Γ interfaces appears to be promising for future device applications.

## Methods

**Crystal growth.** InSe single crystals were grown through the Bridgman method: 6N-pure indium and 5N-pure selenium pellets in an atomic ratio of 52:48 were sealed in an evacuated quartz ampule and subsequently placed into a radio frequency (RF) furnace where the RF power was gradually increased to raise the temperature up to 800 °C. The ampule was then pulled through the hottest zone at a rate of 2 mm/h. Single crystals were characterized via electron dispersive spectroscopy and aberration-corrected transmission electron microscopy. X-ray diffraction measurements confirmed that γ-InSe adopts the R3$m$ space group (160) with unit cell dimensions $a = 4.08(2)$ Å and $c = 24.938(24)$ Å. The quality of the crystals grown in this way is illustrated by the different measurements presented in the main text, including the narrow photoluminescence spectrum measured on exfoliated multilayers encapsulated in between hBN crystals, and the ambipolar transport properties of InSe/TMD heterostructures.

**Sample fabrication.** The fabrication of the heterostructures used to perform the measurements discussed in the main text relies on conventional techniques that are commonly employed to manipulate atomically thin crystals[46] and is briefly outlined here for completeness. Atomically thin layers of TMDs and InSe are obtained by mechanical exfoliation of bulk crystals in a nitrogen gas-filled glove box with a <0.5 ppm concentration of oxygen and water. The exfoliated crystals are transferred onto Si/SiO$_2$ substrates and suitable layers are identified by looking at their optical contrast under an optical microscope. The heterostructures are then assembled in the same glove box with by-now conventional pick-up and release techniques based on either PPC/PDMS (poly(propylene carbonate)/polydimethylsiloxane) or PC/PDMS (polycarbonate) polymer stacks placed on glass slides[46]. To avoid degradation of air-sensitive InSe crystals, the structures are assembled so that the InSe layer lays on a thick hBN layer ($\approx 20$ nm) and is covered by the TMD layer (thereby ensuring that the InSe layer is properly encapsulated in between air-stable materials). Metallic electrodes (Pt/Au) are attached to the TMD layer by conventional nanofabrication techniques using electron-beam lithography, electron-beam evaporation, and lift-off. The sample is wire-bonded with indium or gold wires to a chip carrier and a small amount of ionic liquid ((N,N-diethyl-N-methyl-N-(2-methoxyethyl) ammonium bis(trifluoromethylsulfonyl) imide) commonly referred to as DEME-TFSI) is dropcasted onto the surface of the substrate to cover the metallic gate electrode and the transistor channel. The device is then rapidly transferred into the vacuum chamber with optical access and pumped overnight to remove moisture from the ionic liquid prior to the optical and electrical investigations.

**Optical measurements.** The sample is mounted in a vacuum chamber positioned under an optical microscope, providing optical and electrical access to the sample. The photoluminescence and electroluminescence of our light-emitting field-effect transistors are collected with help of a microscope objective and sent to a Czerny-Turner monochromator with a grating of 150 grooves/mm (Andor Shamrock 500i). The signal is detected with a Silicon Charge Coupled Device (CCD) array (Andor Newton 970 EMCCD). For photoluminescence measurements, the sample is illuminated with a laser beam generated by a supercontinuum white light laser source combined with a contrast filter, allowing to set the illumination wavelength to 610 nm. The laser power is kept at 50 μW, to avoid damaging the structures.

**Transport measurements.** The electrical characterization of our FET is performed in the same chamber used for optical measurements. The gate bias voltage is applied using either a Keithley 2400 source unit or a homemade low-noise voltage source. The current and voltage signals are amplified with homemade low-noise amplifiers, and the amplified signals are recorded with an Agilent 34410A digital multimeter unit.

## Data availability

The data supporting the findings of this study are available free of charges from the Yareta repository of the University of Geneva. https://doi.org/10.26037/yareta:mze42hgmc5cqtdsqmeszkprfxm.

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

## Acknowledgements

We gratefully acknowledge Alexandre Ferreira for continuous and precious technical support. A.F.M. gratefully acknowledges financial support from the Swiss National Science Foundation (Division II) and from the EU Graphene Flagship project. V.F. acknowledges support from EC-FET Quantum Flagship Project 2D-SIPC and EPSRC grant EP/S030719/1. L.B. acknowledges support from US NSF-DMR 1807969 (synthesis, physical characterization, and heterostructure fabrication) and the Office Naval Research DURIP Grant 11997003 (stacking under inert conditions). L.B. acknowledges the use of the facilities at the Platform for the Accelerated Realization, Analysis, and Discovery of Interface Materials (PARADIM), which is supported by the US-NSF under the Cooperative Agreement No. DMR-2039380. D.S. acknowledges support from the U.S. Department of Energy (DE-FG02-07ER46451) for photoluminescence measurements of InSe. D.S. and C.N.L. acknowledge the support by NSF/ECCS award 2128945. The National High Magnetic Field Laboratory acknowledges support from the US-NSF Cooperative agreement Grant number DMR-1644779 and the state of Florida. K.W. and T.T. acknowledge support from the Elemental Strategy Initiative conducted by the MEXT, Japan (Grant Number JPMXP0112101001) and JSPS KAKENHI (Grant Numbers 19H05790, 20H00354, and 21H05233).

## Author contributions

H.H., D.M., D.D., and M.P. fabricated the devices with the help of I.G.L.; H.H. and N.U. measured the devices and analyzed the data. S.M., W.Z., and L.B. grew the InSe crystals. Z.L., D.S., C.N.L., and D.S. performed photoluminescence measurements on hBN encapsulated InSe multilayer, to characterize their quality. V.F. identified the high quality of the InSe crystals and suggested their use the realization of different devices. K.W. and T.T. provided the hBN crystals. H.H., I.G.L., N.U., and A.F.M. discussed the data. A.F.M. wrote the manuscript with input from N.U., H.H., and I.G.L. All the authors read and commented on the manuscript. N.U. and A.F.M. supervised the research.

## Competing interests

The authors declare no competing interests.
