## [Peer Review File · Nature Communications]

Reviewers' Comments:

Reviewer #2:

Remarks to the Author:

I have read the new version submitted by the authors and the revised manuscript. I think the authors satisfactorily addressed my questions. I would like to recommend acceptance.

Reviewer #3:

Remarks to the Author:

The revised manuscript unfortunately does not solve my concern about the novelty worth publication in a high-impact journal.

As noted in Response 3.b, it is true that emission from monolayer TMDs has only one peak and does not change by bias. However, it is not the case for multilayer TMDs. As reported by authors (Figure 6 in *Nano Lett.* 14, 2019), the emission from multilayer TMDs contains both the direct gap and indirect gap recombination with different emission peak energy. The appearance of higher energy emission depends on the energy level because it requires higher doping to reach second band gap (Figure S2 in *Science* 344, 725). That is, the spectral shape depends on the bias. The data in the manuscript shows that several emission paths are switched on one by one during changing the energy level. This behaviour is similar to multilayer TMDs. Authors claim their device as "bias tunable". If so, multilayer TMD devices are bias tunable as well. Hence, the claim in Response 3.c ("van der Waals interfaces are the first tunable light source") is not correct.

Then, what is new of this manuscript is the appearance of another low energy peak due to the interlayer recombination. I do not agree with authors that this point solely guarantee a high-impact journal. I would be happy to recommend consideration for publication in *Nature Communications* if authors can realize a device in which the peak energy of the interlayer recombination can be tuned by bias. Actually, in the manuscript, authors addressed this point as one of the motivations of this work. However, the data shows that the interlayer recombination energy is bias-independent, and then authors ignored this point in the conclusion.

Please find some other issues that would help to improve the manuscript.

1, The thickness of the gating effect (accumulation layer) by ionic gating is about 1nm, less than the thickness of 2L TMDs. How can the gating effect reach the bottom InSe flake that is separated from ions by 2L TMD flakes?

2, Please explain why the EL peak position depends on the thickness of InSe (Figure 4f).

3, To explain the energy difference between low temperature PL and room temperature EL spectra (Figure 3j), authors only refer to the temperature dependent energy gap of TMD (page 4). Temperature dependence of InSe should be considered as well before they can conclude.

4, Authors ascribed the peaks around 1.4 eV (Figures 5b and 5e) to the transition from higher-energy sub-band of the InSe conduction band. However, in their way of operation, judging from Figures 5a and 5d, the doping level of electron side is fixed. Increase of V_{sd} increases the hole doping. Therefore, it sounds weird that increase of V_{sd} leads to the occupation of higher-energy sub-band in the conduction band.

5, In page 6, authors claim "This may account for the peak centered around 1.6 eV in 2L-WS2/4L-InSe, which corresponds well to one of the 2L-WS2 PL peaks." However, this is contradictory to their own data (Figure 3h, blue line).

6, The potential variation shown in Fig. S1 looks different from previous one (Nano Lett. 13, 3023). What is the difference between the two works?

Reviewer #2 (Remarks to the Author):

Comment 2.a: I have read the new version submitted by the authors and the revised manuscript. I think the authors satisfactorily addressed my questions. I would like to recommend acceptance.

Reply 2.a: We are very pleased to see that Reviewer 2 joins Reviewer 1 in recommending our work for publication in Nature Communications. We thank the reviewer for the time spent assessing our manuscript.

Reviewer #3 (Remarks to the Author):

The revised manuscript unfortunately does not solve my concern about the novelty worth publication in a high-impact journal.

We thank the reviewer for the time spent assessing our manuscript. Below we provide our point-by-point replies to the issues raised by the reviewer.

Comment 3.a: As noted in Response 3.b, it is true that emission from monolayer TMDs has only one peak and does not change by bias. However, it is not the case for multilayer TMDs. As reported by authors (Figure 6 in Nano Lett. 14, 2019), the emission from multilayer TMDs contains both the direct gap and indirect gap recombination with different emission peak energy. The appearance of higher energy emission depends on the energy level because it requires higher doping to reach second band gap (Figure S2 in Science 344, 725). That is, the spectral shape depends on the bias.

Reply 3.a: In our reply to the first report of Reviewer 3 we made clear that we know of no previous work on electroluminescent devices based on 2D materials (or, in fact, on any other material platform), in which a broad change in the spectrum of the emitted light could be induced by changing the device operation point (i.e., by changing the source-drain bias and/or the gate voltage). Since the reviewer claimed the contrary, we asked him/her to explicitly cite the papers in which electroluminescence with a light spectrum strongly dependent on bias/gate voltage had been reported. In their new report Reviewer 3 cites two papers and in each one of them he/she refers to a specific figure. *However, contrary to what Reviewer 3 claims, these two papers do not show electroluminescence with a spectrum that strongly depends on bias/gate voltage, neither in the figures mentioned by the referee, nor in any other figure.* In points i) and ii) here below we discuss this in detail.

i) The first reference cited by Reviewer 3 is Jo et al, Nano Lett. 14, 2019 (2014). Two main authors of that paper (Ubrig and Morpurgo) are also main authors of the manuscript currently under consideration. We can exclude with complete certainty that a bias/gate voltage spectrum of electroluminescence is discussed at any point of Nano Lett. 14, 2019 (2014).

In particular, in Fig. 6 (the figure cited by the reviewer) as well as in any other Figure, no panel shows the electroluminescence spectrum measured on the same device for different bias/gate voltages. It is therefore factually not the case that the phenomenon that we discuss in the currently submitted manuscript under consideration had been observed/reported in Nano Lett. 14, 2019 (2014).

ii) The second paper cited by Reviewer 3 is Zhang et al, Science 344, 725 (2014), a reference that the reviewer has already cited in their first report. As we mentioned in our previous reply, the authors of that paper do not claim to have demonstrated the ability to broadly tune the spectral range of the light emitted by their devices, and no figure in that paper shows such an ability. That is: the phenomenon that we discuss in our submitted manuscript has not been reported in that paper (nor claimed to have been observed).

Reviewer 3 cites specifically Fig. S2 (which we reproduce here in Fig. R1 for convenience) and bases their argument on that figure. The main effects that can be observed in Fig. S2 of Science 344, 725 (2014) are that measurements at different bias:

- Show different handedness of the circular polarization;
- Show very strong modulation of the total intensity;
- Show a possible minor variation of the position of the peak in the spectrum by less than 20 meV, much less than the width of the spectral lines shown in the figures.

The data in Fig. S2, therefore, do not show the effect that we are presenting in our manuscript and cannot be used to argue that the phenomena that we report in our manuscript have been reported earlier.

Fig. R1: Figure S2 of the manuscript by Zhang *et al* Science 344, 725 (2014) explicitly mentioned by the reviewer in their report. The electrical biases applied to the device strongly modify the handedness and the intensity of the emitted light, but not (or at most only minorly) the position of the peak. The spectra are obtained at cryogenic temperatures after cooling the sample below the freezing point of the ionic liquid with applied gate biases (see boxes) to freeze a pn-junction in the channel. To change the bias conditions the temperature needs to be cycled and the cooldown repeated under different conditions which is equivalent to realizing a different device in each cooldown. The work by Zhang *et al*, therefore, does not demonstrate an electroluminescent device whose spectrum can be broadly tuned by varying its electrical bias.

At least equally important, the two curves shown in Fig. S2 correspond to measurements performed with *the device that has been cooled down below the freezing temperature of the ionic liquid with different bias and gate voltage in the two cases*. In ionic liquid gate devices, cooling down below the freezing point with different bias conditions creates entirely different electrostatic conditions, effectively corresponding to measuring two different devices. Therefore, not only the curves in Fig. S2 of Science 344, 725 (2014) do not show the effect that we report in our submitted manuscript, but also –if they would have shown the phenomenon– they would not correspond to a simple change of bias and gate voltage, since each measurement has been performed in a different cooldown, done with different bias conditions equivalent to realizing two different devices.

In summary, Fig. S2 of Science 344, 725 (2014) does not report the phenomenon that represents the novelty of our manuscript, namely an electroluminescence signal that can be broadly tuned by simply changing the device bias and gate voltage.

Comment 3.b: The data in the manuscript shows that several emission paths are switched on one by one during changing the energy level. This behaviour is similar to multilayer TMDs. Authors claim their device as “bias tunable”. If so, multilayer TMD devices are bias tunable as well.

Reply 3.b: It may be that the phenomenon that we report may also happen in multilayer TMDs, because the basic ingredients on which the phenomenon relies –the presence of multiple subbands/valleys and a potential difference between different layers– are present in TMD multilayers as well. Indeed, we do argue in our manuscript that some of the peaks that we see in our electroluminescence spectra may originate from higher energy intralayer transitions. However, so far nobody reported the realization of any device in which this phenomenon is observed, and even less to realize a light source with a bias tunable spectrum. As we have repeatedly claimed, we do not know of any published work reporting such a phenomenon, and the papers that the Reviewer cited (after that we have explicitly asked him/her to produce references to support their claims) also do not show it.

We therefore maintain that the manuscript that we have submitted represent the first report of electroluminescent devices exhibiting a broadly tunable light spectrum upon changing bias and gate voltage.

Comment 3.c: Hence, the claim in Response 3.c (“van der Waals interfaces are the first tunable light source”) is not correct.

Reply 3.c: For the reasons explained in the replies here above, we maintain that the main claim of novelty that we make in our manuscript is fully correct and valid. Light-emitting transistors based on Γ - Γ van der Waals interfaces are the first reported light sources whose spectrum can be broadly tuned by changing bias and gate voltage. In their two reports, Reviewer 3 has not produced any evidence to the contrary.

Comment 3.d: Then, what is new of this manuscript is the appearance of another low energy peak due to the interlayer recombination. I do not agree with authors that this point solely guarantee a high-impact journal. I would be happy to recommend consideration for publication in Nature Communications if authors can realize a device in which the peak energy of the interlayer recombination can be tuned by bias. Actually, in the manuscript, authors addressed this point as one of the motivations of this work. However, the data shows that the interlayer recombination energy is bias-independent, and then authors ignored this point in the conclusion.

Reply 3.d: Reviewer 3 writes explicitly that he would recommend publication if our devices would show light emission due to interlayer transitions that can be tuned by changing the bias. Reviewer 3 should then realize that we already show this result in our manuscript, as it is apparent, for instance, in Fig. 6a and 6b of the main text (which we reproduce here in Fig. R2 below for convenience).

The device whose data are shown in these figures is formed by a 4L-InSe and a 2L-WS₂, and we know well the energy of the intralayer radiative transitions in the two materials. The lowest intralayer transition energies in InSe and WS₂ are respectively at 1.55 eV and 1.65 eV. Therefore, all emission lines at energy below 1.55 eV are necessarily originating from interlayer transitions. For convenience, we draw the vertical dashed red lines in the Figures here at 1.55 eV.

It is clear that upon changing V_G , the emission line changes by as much as 300 meV: at large positive V_G (between 0.4 and 0.8 V) the emission energy is 1.2 eV independently of V_G , whereas for V_G between -0.1

V and 0.2 V, the position of emission line shifts from just under 1.3 eV to above 1.5 eV. We can then conclude directly from the experimental data that –by doing nothing else than changing V_G – our devices allow tuning the spectrum of light emitted by interlayer transitions between 1.2 eV and 1.5 eV. A variation of 0.3 eV in the interval 1.2-1.5 eV represents a very broad range of spectral tunability (and if beyond interlayer transitions, we also consider the effect of intralayer ones, the range of spectral tunability is even larger).

Note that in the caption of Fig. 6 of the original version of our manuscript we did write explicitly “...; for $V_G < 0.2$ V a broader peak centered around 1.4 eV and shifting with varying V_G is observed”. That is: we did remark explicitly on the fact that we observe the phenomenon that Reviewer 3 wants us to report.

Possibly, Reviewer 3 did not appreciate the fact that we are already tuning the energy of interlayer transitions by a large amount, because he/she may have expected the energy of the emitted light to vary linearly with V_G over the entire V_G range of the experiments. There is however no reason to expect that, because of large and non-linear screening effects, which affect the p-n junction region in a spatially dependent way. Indeed, on one side of the p-n junctions there are electrons in the bottom InSe layer, on the other side there are holes in the top WS_2 layer, and in the recombination region the potential difference between the two layers is strongly dependent on position, with details that depend on the device operation point. *Our conclusion that the spectrum of the emitted light can be tuned broadly, however, does not rely on these details and can be drawn directly from the experimental observations, simply because we know in which energy range light is necessarily emitted by interlayer transitions. That is why our claim is robust.*

Fig. R2: **a.** Color plot of the intensity of the light emitted by a 2L- WS_2 /4L-InSe LEFET device as a function of photon energy and gate voltage (V_{SD} is fixed to +2.4 V). The vertical red dashed line delimits the lowest intralayer transition energies. All emission lines at energies below 1.55 eV are necessarily originating from interlayer transitions. The data show that varying V_G allows tuning the spectrum of light emitted by interlayer transitions between 1.2 eV and 1.5 eV. **b.** Individual horizontal cuts of the color plot shown in a, for V_G varying from -0.1 V to +0.7 V in 0.1 V steps.

Comment 3.e: Please find some other issues that would help to improve the manuscript.

1, The thickness of the gating effect (accumulation layer) by ionic gating is about 1nm, less than the thickness of 2L TMDs. How can the gating effect reach the bottom InSe flake that is separated from ions by 2L TMD flakes?

Reply 3.e: Electrostatic screening is certainly present but does not prevent tuning the spectrum of the emitted light by changing the potential in the two layers, which determine the energy of interlayer transitions. There are multiple reasons, whose relevance may depend on the specific operating regime of the device.

The most important reason is that light originates from the recombination of electrons and holes in different layers, taking place in the pn-junction region, which has a finite lateral extension. One side of the pn-junction hosts holes in the top layer (which can screen the ionic gate on top thereby suppressing electrostatically the potential difference between the top layers) and the other side electrons in the bottom layer (which do not screen the ionic gate). As we move from one side to the other the density of holes decreases and that of electron increases (see scheme in Fig. R3), changing the screening locally, and changing the sensitivity of the transition energy to the electric field associated to the applied gate voltage. As a result, at different positions in the pn-junction the electrostatic potential difference between the two layers varies, so that the difference between the electron and hole states –the two states involved in the radiative transition– varies as well. As a result, the energy of the emitted photons depends on the position in the pn-junction at which the transition occurs, and the sensitivity to the gate voltage depends on the precise configuration of charge density in the two layers. One of the consequences of this situation is a relatively large spread in recombination energies, which results in a broad linewidth, as effectively observed in the measurements.

Fig. R3: Schematics of the charge distribution in our LEFET devices in the ambipolar injection regime. The top panel shows a side view of the device with the top layer (TMD) hosting holes on the left side and with electrons accumulated in InSe in the bottom layer, on the other side. The curves on the bottom panel show the density of holes (red, n_h) and electrons (blue, n_e) along the channel between source and drain contact (the curves are meant to illustrate qualitatively the behavior). Across the pn-area, where the recombination process takes place, the density of holes decreases and that of electrons increases when moving from left to right, resulting in a variation of the local screening and a spatially dependent electrostatic potential.

A quantitative description of screening in such a system is obviously complex, and the complexity is increased by the fact that the applied source drain voltage not only affects the potential in the two layers, but also injects both holes and electrons with very high energy, since source-drain biases of several Volts are applied to reach the ambipolar injection regime. Such high injection energy creates a distribution of charge that is strongly out of equilibrium and that cannot be simply described by an electrochemical potential level, as it is the case for the system in equilibrium.

That is why—in view of this complexity—in our manuscript we emphasize all conclusions that can be drawn from the experimental data without a detailed knowledge of the electrostatics of the system. This is possible, because interlayer transitions can be identified from their energy (lower than for intralayer transitions) and detecting the change in the spectrum upon changing bias and gate voltage can be done directly by looking at the measurements. This is all we need to draw our conclusions.

Comment 3.f: 2, Please explain why the EL peak position depends on the thickness of InSe (Figure 4f).

Reply 3.f: The dependence of the interlayer transition energy as a function of InSe and WS₂ thickness has been addressed in detail in Ubrig et al., Nat. Materials. (2019). The emission energy of the interlayer transition is determined by the relative alignment of the conduction and valence band edges in the two materials. Since the band gaps of InSe and WS₂ multilayers depend on thickness the distance in energy of the conduction and valence band edge in the two materials also depends on thickness, and that is the reason for the thickness dependence of the interlayer transition energy.

Comment 3.g: 3, To explain the energy difference between low temperature PL and room temperature EL spectra (Figure 3j), authors only refer to the temperature dependent energy gap of TMD (page 4). Temperature dependence of InSe should be considered as well before they can conclude.

Reply 3.g: The bandgap of InSe, like for most semiconductors, increases upon lowering temperature. The bandgap of InSe has been shown to increase by approximately 50 meV upon cooling from 300 K to 4 K (see Shubina et al, Nature communications 10, 3479 (2019), Patil et al emergent mater. 4, 1029-1036 (2021)). Taking this into account and the increase of gap for both semiconductors, the total increase of the interlayer transition energy upon cooling is expected to be approximately 100-150 meV, consistently with what we observe in the experiments. We have added references in the main text to avoid any confusion.

Comment 3.h: 4, Authors ascribed the peaks around 1.4 eV (Figures 5b and 5e) to the transition from higher-energy sub-band of the InSe conduction band. However, in their way of operation, judging from Figures 5a and 5d, the doping level of electron side is fixed. Increase of V_{sd} increases the hole doping. Therefore, it sounds weird that increase of V_{sd} leads to the occupation of higher-energy sub-band in the conduction band.

Reply 3.h: Increasing the source-drain has two main effects in the ambipolar injection regime. First, the density of electrons and holes changes throughout the channel. Second, the energy with which both electrons and holes are injected into the channel increases (both electrons and holes are injected at high energy in the ambipolar injection regime, not only holes as the reviewer writes). As source-drain bias of several volts are applied in the ambipolar injection regime, this generates a population of electrons and holes that is very strongly out of equilibrium, which cannot be described in terms of an electrochemical potential (the non-equilibrium distribution can deviate substantially from a Fermi-Dirac distribution). Under such conditions, both electrons and holes occupy high energy states (i.e., states in higher energy sub-bands).

Comment 3.i: 5, In page 6, authors claim “This may account for the peak centered around 1.6 eV in 2L-WS₂/4L-InSe, which corresponds well to one of the 2L-WS₂ PL peaks.” However, this is contradictory to their own data (Figure 3h, blue line).

Reply 3.i: The 2L WS₂ peak in EL at 1.6 eV is fully consistent with our interpretation. The difference noted by the referee originates from the fact that in Figure 3h all spectra are measured at low temperature, since at room temperature PL is either weak or absent. At room temperature, the Γ -K transition in 2L WS₂ occurs at 1.6 eV.

Comment 3.j: 6, The potential variation shown in Fig. S1 looks different from previous one (Nano Lett. 13, 3023). What is the difference between the two works?

Reply 3.j: What we write in the supplementary material is meant to explain qualitatively the origin of the ambipolar injection regime. Such a simple qualitative explanation is useful for readers unfamiliar with ionic gated transistors, and is used in textbooks and review articles --see for instance *Sze, Ng, Physics of Semiconductor Devices* or Kang and Frisbie ChemPhysChem 14, 1547 (2013). In contrast, Zhang *et al* (Nano Lett. 13, 3023) try to give a more detailed and accurate description of the potential profile in the ambipolar injection regime, where the potential drops sharply at the pn interface. For the interested reader we have added the reference given by the reviewer to the supplementary information.

Reviewers' Comments:

Reviewer #1:

Remarks to the Author:

In my view, the assignment of the low energy EL peak to interlayer excitons is reasonable given recent progress on understanding PL and EL properties of such heterostructures. Although more studied could have been done to probe the properties of such excitons, I do feel that the assumption is reasonable and consistent with current understanding. With that, I would support acceptance of the revised manuscript based on the significance of the bias-tunable interlayer exciton EL.

Reviewer #2:

Remarks to the Author:

The referee and the authors argue about the emission mechanisms and the novelty of the work. In terms of physical mechanisms, I would say both parties make their points. In terms of technological advancement, I do not agree with the statement made by the authors "However, so far nobody reported the realization of any device in which this phenomenon is observed, and even less to realize a light source with a bias tunable spectrum". Bias-tunable light sources made of 2D materials have been demonstrated in previous reports (not only those papers cited in this manuscript 49-52 but there are many other papers that are not included, e.g., Nat. Commun. 10, 1709 (2019). Broadband tunable EL emission in 2D materials is also achieved in recent work, e.g., Nature 596, 7871, 232-237 (2021).

I think the novelties of the submitted manuscript are 1) the EL emission from a "vdW heterojunction" which is 0.2 eV-tunable by the gate and 2) the e and h injection scheme enabled by the ionic liquid gate. The first one is an interesting finding, but the authors and I both agree that the emission efficiency would be extremely low due to the inefficient indirect interband transition. This finding indeed has some scientific values but is likely not applicable. The second one is refused by the authors themselves as they claimed it has been reported before.

Overall, the physics is valid but the novelty is moderate. I would like to recommend consideration for publication depending on whether referee #3 is satisfied with the reply in Comment 3.d.

Reviewer #3:

Remarks to the Author:

My main concern was not addressed properly. It seems authors have interpreted my comment by their own when they prepare replies.

First of all, I did not mention in my first report that in their previous work (Nano Lett. 14, 2019) electroluminescence was bias tunable. I referred to this journal only as an example that bilayer TMD has multiple electroluminescence peak. Regarding the second paper (Science 344, 725), authors actually agree by themselves in their response that "*The main effects that can be observed in Fig. S2 of Science 344, 725 (2014) are that measurements at different bias... - Show very strong modulation of the total intensity...*". In the present manuscript, most of the data shows that the intensity of each emission peak is strongly modulated by differing the bias. Therefore, their criticism "*However, contrary to what Reviewer 3 claims, these two papers do not show electroluminescence with a spectrum that strongly depends on bias/gate voltage*" is completely misdirected.

Authors explain that "*In ionic liquid gate devices, cooling down below the freezing point with different bias conditions creates entirely different electrostatic conditions, effectively corresponding to measuring two different devices.*" I know that different bias creates different electrostatic condition, irrespective of the temperature, but so does the device in the present manuscript. "Changing the operating point" mentioned by authors in the present manuscript is nothing but changing the bias condition. Authors cannot claim that the device in previous work is effectively

different while their device in the present manuscript is identical.

Similarly in Reply 3.c, I appreciate that the present manuscript is the first report of the electroluminescence from Γ - Γ van der Waals interfaces. I only mentioned that it is not the first "bias tunable" device. Even though previous work (Science 344, 725) did not mention the bias tunability, their data shows it. This is mentioned in the response letter as well, as I pointed out above. Again, the authors' criticism "*In their two reports, Reviewer 3 has not produced any evidence to the contrary*" is misoriented. Authors already agreed by themselves that I had supplied evidence.

Regarding Reply 3.d, I do know that there is a signal that their device shows a "really" bias tunable operation, which requires a van der Waals heterostructure. In my opinion, this is the only signature that matches the title and may worth the standard of the journal with a sophisticated study. However, at the present stage, data and understanding are too premature. In the first manuscript, authors only "tentatively" ascribed this signal to the interlayer transition. I think this was honest and scientific based on the premature results. However, following my report, they revised the main text to claim it IS the interlayer transition without further investigation. If authors wish not to devote their effort to improve the manuscript, then I would like not to change my attitude: I cannot recommend publication.